# Endometriosis and Opioid Receptors: Are Opioids a Possible/Promising Treatment for Endometriosis?

**DOI:** 10.3390/ijms24021633

**Published:** 2023-01-13

**Authors:** Qihui Guan, Renata Voltolini Velho, Jalid Sehouli, Sylvia Mechsner

**Affiliations:** Department of Gynecology Charité with Center of Oncological Surgery, Endometriosis Research Center Charité, Campus Virchow-Klinikum, Augustenburger Platz 1, 13353 Berlin, Germany

**Keywords:** chronic pelvic pain, drug abuse, NOP receptor, opioid tolerance, opium, therapeutic option

## Abstract

Endometriosis (EM), defined as the presence of endometrial-like tissue with surrounding smooth muscle cells outside the uterus, is a disregarded gynecological disease reported to affect 6–10% of women of reproductive age, with 30–50% of them suffering from chronic pelvic pain and infertility. Since the exact pathogenic mechanisms of EM are still unclear, no curative therapy is available. As pain is an important factor in EM, optimal analgesia should be sought, which to date has been treated primarily with non-steroidal anti-inflammatory drugs (NSAIDs), metamizole or, in extreme cases, opioids. Here, we review the pain therapy options, the mechanisms of pain development in EM, the endogenous opioid system and pain, as well as the opioid receptors and EM-associated pain. We also explore the drug abuse and addiction to opioids and the possible use of NOP receptors in terms of analgesia and improved tolerability as a target for EM-associated pain treatment. Emerging evidence has shown a promising functional profile of bifunctional NOP/MOP partial agonists as safe and nonaddictive analgesics. However, until now, the role of NOP receptors in EM has not been investigated. This review offers a thought which still needs further investigation but may provide potential options for relieving EM-associated pain.

## 1. Introduction

Endometriosis (EM), defined as the presence of endometrial-like tissue with surrounding smooth muscle cells outside the uterus, is a disregarded gynecological disease reported to affect 6–10% of women of reproductive age and 30–50% of them suffering from chronic pelvic pain and infertility [1,2,3]. Patients complain that pain appears within the reproductive and urinary system, or lower part of the digestive system, which is related to the location of EM focus [4,5,6]. Furthermore, typical EM symptoms include dyspareunia, dysmenorrhea, dysuria, dyschezia, and noncyclic chronic pelvic pain [7]. Some patients also complain of atypical symptoms, such as pain in the lumbar-sacral region of the spine, diarrhea, constipation, fatigue, flatulence, nausea, anxiety, depression, and headaches, which seem to be stress-induced [8]. All these symptoms and a long course of the disease lead to a significant decrease in patient quality of life [9].

Since the exact pathogenetic mechanisms of EM are still unclear, no curative therapy is available. The treatment goal is to alleviate the symptoms and prevent the further spread of EM while preserving fertility [3,10]. The first line of therapy includes combined oral contraceptives or progestins. The second line includes gonadotropin-releasing hormone analogues (GnRHa) and aromatase inhibitors [11]. Since these therapies interfere with ovulation, the desire for an active or future pregnancy must be considered when prescribing them [12]. Moreover, about 30–60% of EM patients do not respond to conventional treatment [13,14]. In addition, about 10% of patients cannot continue the treatment because of the adverse effects (vaginal dryness, decreased libido, depression, irritability, fatigue, bone mineral loss, changes in lipid profile, weight gain, oedema, acne, hot flashes, liver toxicity, breast atrophy, voice alteration, hirsutism, and oily skin) inherent to the therapy [15,16,17]. Moreover, a systematic review was able to show that the various drugs do not differ significantly in terms of their pain-relieving effect [18]. Although the drugs can prevent the further progression of the disease, existing endometriotic structures remain, regardless of the specific drug used, the dosage, or the duration of use [10]. Therefore, EM-related symptoms can reappear when drug use is terminated or interrupted, which is why drug therapy is regarded as a long-term treatment.

As the third line of treatment, the macroscopically visible EM foci can be removed by a surgical procedure (laparoscopy). Patients experiencing symptoms including severe involvement of the ovaries or ovarian cysts, stenosis in the bladder or gastrointestinal tract caused by EM, pain resistant to therapy, or a unfulfilled desire to have children are recommended for this procedure [19]. Our group has showed that 23.8% of patients had no pain after laparoscopy, while 52% had an improvement in symptoms [20]. Although this intervention seems to be a successful treatment option, pain remains in 25% of patients and EM recurrence is between 40 and 45% [14]. However, independent of the treatment chosen, the most important point is the early intervention to avoid pain chronification. Given that most adult patients report symptoms of EM as well as CPP during adolescence, and that approximately 80% of adolescent girls with CPP not responding to conventional medical therapy have EM, this would be a critical time to intervene [21].

As pain is an important factor in EM, optimal analgesia should be sought, which to date has been treated primarily with non-steroidal anti-inflammatory drugs (NSAIDs), metamizole or, in extreme cases, opioids [3]. However, the response to NSAIDs is often ineffective [22]. In addition, NSAIDs are associated with a higher risk of gastrointestinal bleeding. As for metamizole, agranulocytosis is the most notorious adverse event, as well as metamizole-associated hepatotoxicity [23]. Opioids should be prescribed but strongly supervised, as there is a risk of addiction [24].

Here, we review the mechanisms of pain development in EM, the endogenous opioid system and pain, as well as the opioid receptors and EM-associated pain. We also explore the drug abuse and addiction to opioids and the possible use of NOP receptors in terms of analgesia and improved tolerability as a target for EM-associated pain treatment. This review offers a thought which still needs further investigation, but may provide potential options for relieving EM-associated pain.

## 2. Mechanisms of Pain Development in Endometriosis

Mechanisms that lead to the development of pain in EM are very complex and multi-layered. Peripheral pain generation mechanisms that take place at the local level can be distinguished from those that take place in the central nervous system. Regionally, nociceptors are activated in and around the endometriotic lesions, which convert the stimulus into action potentials (Figure 1). These are in turn transmitted via the spinal cord to the brain, where signal processing takes place. This results in the subjective perception of pain [25]. Nociceptors are sensory receptors activated by noxious stimuli, which in EM represent pain mediator release and nerve infiltration or compression, but that is not usual. A pain receptor involved in this mechanism is the transient receptor potential vanilloid channel 1 (TRPV1), which acts as a cation channel and is increased in EM patients [26,27]. When the pain originates in the parietal peritoneum, it is described as a well-localized somatic pain. Visceral pain results from the activation of nociceptors in organs such as the uterus, bladder, or rectum. Compared to somatic pain, visceral pain tends to be dull and difficult to localize [7].

Inflammatory pain is also a subtype of nociceptive pain [28,29]. EM is associated with a local inflammatory reaction in which regional inflammatory mediators such as cytokines, tumor necrosis factor-alpha (TNF-α), interleukin (IL)-1β, IL-6, and also immune cells are present in high concentrations [10,30]. The inflammatory mediators are secreted both by the endometriotic lesions themselves and by surrounding cells and nerve tissue [25]. The resulting neurogenic inflammation leads to hypersensitivity and activation of pain receptors [31], which is often associated with regional hyperalgesia (increased local pain perception) [12,30]. This is the result of longstanding chronic inflammation in which pain loses its function as a barrier to noxious stimuli [32]. In addition, neuropathic pain is described in up to 75% of EM patients. Neuropathic pain is defined as pain caused by a lesion or disease of the somatosensory nervous system. The peripheral mechanisms of EM-associated pain are mainly focused on the interplay among the immune system, peripheral nerve system, and endometriotic lesions [33]. In the case of EM, the nerves in the damaged area are activated without an active stimulus and patients often perceive it as a sharp pain [25].

After the pain is perceived via the peripheral anatomical area, the consciousness of pain emerges from the central nervous system (CNS) [34]. The modulation of chronic pelvic pain in women with EM and EM-associated pain correlates with the CNS [35,36]. Neural mechanisms similar to the generation of memory may be why central sensitization can cause pain without a peripheral noxious input [37]. Factors such as mood, feelings, and previous pain experiences significantly influence pain processing and can lead to both reduced and increased pain perception (centralized pain). Therefore, there is no linear relationship between the perceived pain intensity and the strength of the stimulus [25]. For example, women who suffer from dysmenorrhea react more sensitively to painful noxae, such as heat, than women without menstrual pain. In addition to the subjectively perceived stronger pain, an increased pain response in the brain was also shown in dysmenorrhea patients. Dysmenorrhea is a risk factor for chronic pelvic pain [38].

Additionally, one widely accepted view is that nociceptive pathways change in an activity-dependent manner, i.e., show plasticity [39,40,41,42,43]. The concept of functional plasticity or neuroplasticity provides mechanistic links between specific changes in molecules, synapses, microcircuits, and systems and thereby links a variety of modulatory factors to a change in perception and behavior. Over the last three to four decades, studies in animal models of chronic pain have established that peripheral afferents sensitize in response to a variety of molecules secreted by nonneuronal cells (immune cells and blood vessels, including inflammatory cytokines and growth factors) [41].

Increasing evidence has led to EM being considered as a neurogenic inflammatory disease [27,30]. Elevated expression and activation of nociceptors and elevated levels of neuropeptides, other proinflammatory chemicals, and cytokines imply that neuroinflammatory processes are present in the CNS in EM [30]. Although the precise mechanism of how pain in EM generates is unknown, it is clear that adequate analgesia is an integral and important part of EM treatment.

## 3. Endogenous Opioid System and Pain

The endogenous opioid system is integrated with endogenous opioid peptides and receptors [44]. The opioid receptors are transmembrane proteins belonging to the seven transmembrane-spanning superfamily of G-protein-coupled receptors (GPCR). GPCRs are of fundamental physiological importance, mediating actions of the majority of known neurotransmitters and hormones. Binding studies and bioassays defined four main types of opioid receptors: delta (DOR), mu (MOR), kappa (KOR) opioid receptor, and nociceptin/orphanin FQ peptide (NOP, initially called LC132, ORL-1) receptor [44,45].

The endogenous opioid system plays a fundamental role in modulating neurogenic inflammation and the ensuing pain and is implicated in the physiological control of emotional and cognitive responses [46]. Corticotropin-releasing factors, cytokines, catecholamines, and environmental stimuli such as stress can liberate endogenously occurring opioid peptides (e.g., β-endorphin) [47]. These activate the neuronal opioid receptors leading to the inhibition of the excitability of these nerves or the release of proinflammatory neuropeptides that results in analgesia [48]. Analgesic effects of locally administered opioids are particularly prominent under inflammatory conditions. Studies have shown that there may be an inflammation-dependent upregulation of opioid receptors in the periphery. After induction of peripheral inflammation, axonal transport of opioid receptors to the nerve endings will enhance, which leads to an increase in their density on peripheral nerve terminals [49,50]. In inflamed tissue, mRNAs for β-endorphin and enkephalins were found in lymphocytes, monocytes, and macrophages. Immune cells express the required machinery to process proopiomelanocortin into β-endorphin and release it from secretory granules [51].

## 4. Opioid Receptors and EM-Associated Pain

Since EM is a chronic inflammatory disease with disturbing pain mediation and analgesia, opioid system disruption (opioid peptides and/or receptors) might be involved in the inflammatory condition and pain pathogenesis. Indeed, some studies have focused on opioid receptor expression in the endometrium (Figure 2) [52,53,54,55,56].

Among the opioid receptors, KOR plays an important role in visceral and inflammatory pain [57,58] and its stimulation produces greater analgesia in women than in men due to sex differences associated with κ-opioid agonism [59]. Moreover, kappa-opioid receptor expression has been described in ectopic endometrial tissues of female rodents and women [53,60,61]. A recent study has shown that KOR stimulation can alleviate and prevent chronic pelvic mechanical sensitivity and discomfort in female mice subjected to EM. This KOR-mediated pain relief was void of antiallodynic tolerance and was highly effective during estrus, the phase of the estrous cycle in which mice become more sensitive. Such an increased sensitivity resembles the intense perimenstrual pain observed in EM patients. Interestingly, KOR-mediated pain relief did not modify the anxiety-like behavior or the memory impairment of mice with ectopic endometrial growths [46]. Kappa-opioid receptor agonists showed analgesia in patients with non-ulcer dyspepsia [62] and irritable bowel syndrome [63,64].

MOR is expressed in the human endometrium and its expression pattern changes during the menstrual cycle differently in all endometrial compartments. These findings suggest that MOR could have several functions in the complex remodeling process that the endometrium undergoes every month and, therefore, in EM [56]. The mu-opioid receptor mRNA was upregulated in EM stromal cells, indicating that it may be involved in a defective immune system in this disease [65]. Additionally, MOR expression was significantly higher in ovarian EM than in eutopic endometrium [66]. Mu-opioid receptor expression was not detected in stromal cells from GnRH agonist-treated patients with deep infiltrating EM (DIE). Although MOR expression in stromal cells was detectable in progestin-treated patients, the expression levels were significantly lower than those in untreated patients [55]. These findings suggest that neuroimmune interactions may play a crucial role in the pathogenesis of EM-associated pain. Mediators including MOR in the neuroimmune pathways may be new targets for non-hormonal treatments.

In an EM rat model, a decrease in MOR immunoreactivity within neuronal compartments was reported [67]. This study demonstrated that the EM impact on the periaqueductal grey (PAG) opioid system is related to changes in MOR and N-methyl-D-aspartate receptor (NR1) subunit expressions. It indicated that EM might influence opioidergic and glutamatergic activities in the PAG. In addition, studies showed that there may be an interaction between psychological stress and EM development and progression [68]. EM-associated pain may be alleviated after physical interventions and aggravated under psychological stress [69,70].

The discovery of *OPRD1*, which is the DOR encoding gene, began an era of molecular and genetic investigations of the opioid system. *OPRD1*-knockout mice revealed that DORs have anxiolytic and antidepressant functions [71], decidedly distinguishing this receptor from MORs. Many pharmacological studies have shown DORs in mood disorders and chronic pain [72,73,74]. Other researchers have reported that DOR has great potential for the treatment of chronic pain with ancillary anxiolytic- and antidepressant-like effects [73,75,76]. DOR agonists were reported specifically active in the periphery, and may significantly improve the treatment of chronic inflammatory visceral pain [77]. The DOR is widely spread in the brain, synthesized in primary afferents and transported to the spinal cord [78]. The precise distribution of DOR in dorsal root (DRG) neurons remains controversial. Some studies infer that DOR is mainly expressed in large myelinated DRG neurons and shows a low level of co-expression with MOP [79,80]. Others have indicated that DOR is also distributed in small DRG neurons and that DOR and MOR can inhibit the noxious heat and mechanical-induced release of SP in the spinal cord [81,82,83]. However, to date, there is no evidence showing that EM-associated pain can be relieved by DOR antagonists.

In the mid-1990s, nociceptin/orphanin FQ (N/OFQ) was identified as a multifunctional ligand for the opioid receptor-like 1 (ORL1), with ORL1 renamed to N/OFQ peptide (NOP) receptor [84]. Despite the high homology of the NOP receptor and other classical opioid receptors (MOP, DOP, and KOP), N/OFQ does not bind to classical opioid receptors owing to its unique structure [85]. Both the NOP receptor and the endogenous ligand N/OFQ are widely expressed both in the central and peripheral nervous systems, as well as in peripheral organs, such as the heart and intestines, and the immune system of rodents and humans [85]. Given their distribution, N/OFQ and NOP receptors contribute to the regulation of different functions such as memory, emotion, reward, motor function, and sensory processing. Moreover, they are related to the regulation of renal and cardiovascular functions, respiratory functions, cough reflexes, urinary bladder function, and micturition reflexes, with special regard to sympathetic and parasympathetic regulation [86,87,88]. Moreover, its involvement in pain modulation has been reported [89]. Spinal NOP receptor activation produces anti-hyperalgesic and anti-allodynic effects in chronic pain [89].

In animal models, intrathecal N/OFQ administration inhibited thermal hyperalgesia of chronic inflammatory pain [90,91], and similar effects were observed in neuropathic pain models caused by chronic constriction injury (CCI) or spinal nerve ligation [90]. Compared with the other classical opioid receptors (MOR, KOR, and DOR), NOP receptors can be detected and their expression was unaffected by opioids and reduced by LPS/PepG combinations, while the aforementioned three classical opioid receptors could not be detected [92]. This study highly indicated that classical opioid receptors are not expressed in circulating immune cells. Indeed, NOP receptors have a high distribution within the immune system, including in lymphocytes, monocytes, B/T cells, and mononuclear cells [93,94,95].

There is a strong correlation between the plasmatic N/OFQ levels and the severity of sepsis [96,97,98], Parkinson’s disease [99,100,101], arthritis [102], and inflammatory bowel diseases [103,104,105]. Other researchers have reported NOP receptors as a novel potential target in the treatment of gastrointestinal diseases, including inflammatory bowel diseases and irritable bowel syndrome [106,107]. In addition, its therapeutic potential in the treatment of traumatic brain injuries, traumatic stress, and their co-morbidities has also been reported [108].

Everything considered, it makes sense to believe that NOP may play a crucial role in alleviating EM-associated pain via the anti-neuroinflammatory process. However, to date, the relationship between EM-associated pain and NOP or N/OFQ remains unknown.

## 5. Opioids and Female Reproduction

The presence of the opioid system in peripheral reproductive tissues in both women and men is a recent revelation [109]. Specifically, researchers have reported the expression of opioid receptors in the granulosa cells of the ovarian follicle, in the oocyte, and in the human endometrium [56,110,111]. Opioid receptors are concurrently found in the somatic and germ cells of the testis. The clinical correlations of these findings include effects on follicular maturation, embryo implantation, spermatogenesis, and sexual function [109].

In the female central nervous system, they are involved in controlling the release of GnRH and thus the gonadotropins luteinizing hormone (LH) and follicle-stimulating hormone (FSH). This affects ovarian hormonal production, follicular growth, and ovulation. Peripherally, endogenous opioids act directly as important neuromodulators as well as signaling peptides within several reproductive organs and tissues. These include the endocrine pancreas, the different compartments of the ovary, including the ovarian follicles and the oocyte, as well as the endometrium. The endogenous opioid system also has an important role during pregnancy and parturition. In addition to its direct roles, opioids also affect the prolactin and oxytocin systems, exerting additional indirect effects [109].

An interaction between opioid tone, GnRH secretion and subsequent release of LH and FSH has been proposed regarding to the menstrual cycle [112]. In the early follicular phase, the opioid tone is low, resulting in an uninhibited pulsatile GnRH secretion with a high frequency. The rise in estradiol levels towards the midfollicular phase is accompanied by a rise in the opioid tone. In turn, the rise in opioids promotes slower GnRH pulses leading to an increase in LH pulse amplitude. The frequency of GnRH and LH pulses is reduced in the luteal phase, caused by a high opioid tone induced through high progesterone levels [109]. A study with 131 healthy volunteers corroborated the postulated interaction, in which it was shown that β-endorphin levels, one of the endogenous opioid peptides which mainly act on µ-opioid receptors, increase during the follicular phase, reaching a maximum approximately 4 days before ovulation. The β-endorphin levels further increase in the luteal phase, before decreasing again before the next menstrual bleed. A positive linear correlation between endogenous opioids and progesterone levels during the luteal phase has been found [113].

The effect of opioids on the hypothalamic–pituitary axis (HPA) depends on the phase of the menstrual cycle. In ovariectomized rats, a high dose (10 mg/kg) of morphine increased LH but decreased it in low doses (1 mg/kg), which provides evidence that the effect of a single injection of morphine on LH is dose-dependent [114]. The primary mechanism by which opioids affect gonadotropin secretion is through their effects on GnRH. By in situ hybridization, GnRH mRNA levels are downregulated by opioids, suggesting that the suppression of the biosynthesis of GnRH may be stimulated by morphine [115]. Opioids have an effect on steroids via the suppression of gonadotropins. Gonadal steroids, in turn, decrease the effects of opioids on LH secretion. Opioids also play an important role in the feedback inhibition of LH by gonadal steroids. In female rats, both estradiol-mediated negative feedback and the estradiol surge-induced hypersecretion of LH are upregulated by chronic opioid treatment, suggesting that opioids amplify negative and positive feedback on gonadotropin secretion [116]. Compared with saline controls, opioid antagonists promoted an increase in the LH surge [117].

A potential relationship between long-term opioid use and reduced libido, hypogonadism, and reproductive dysfunction in women has been reported [118,119,120,121,122,123]. A study that compared 68 non-opioid-consuming women control subjects with 47 women (aged 30–75 years) who were consuming sustained action oral or transdermal opioids for control of nonmalignant pain found that testosterone, estradiol, and dehydroepiandrosterone sulfate values were 48% to 57% lower in opioid-consuming women with intact ovarian tissue than in the control subjects. LH and FSH values were on average 30% lower in premenopausal and 70% lower in postmenopausal opioid consumers. Among oophorectomized women not consuming estrogen, free testosterone levels were 39% lower in opioid consumers, indicating impaired adrenal androgen production. Additional lowering of free testosterone levels was associated independently with oral estrogen replacement and low body mass index. Menstruation had often ceased soon after beginning sustained action opioid therapy. These results document hypogonadotropic hypogonadism and decreased adrenal androgen production in most women consuming sustained action oral or transdermal opioids [124]. In support of these results, Rhodin et al. [119] found that in the opioid-treated group, the patients had signs of pituitary dysfunction affecting all axes. Significant differences were shown in hypofunction of the hypothalamic–pituitary–gonadal axis and hyperfunction of the hypothalamic–pituitary–adrenal axis, and higher prolactin levels were found in the opioid-treated group compared with the control group. The degree of pain was rated the same in both groups, but the opioid-treated group reported more side effects and a lower quality of life.

Taken together, future research into the role of endogenous opioids in reproductive disorders may lead to a better understanding of the pathophysiology of reproduction as well as to novel treatments options.

## 6. Drug Abuse and Addiction to Opioids

Opium, extracted from the seeds of Papaversomniferum, has been known for millennia to relieve pain, and its use for surgical analgesia has been recorded for several centuries. Most likely, opium was the first narcotic substance discovered at the dawn of humankind. The history of drug addiction is immensely rich and allows to trace the long way that humankind has had to travel to reach the contemporary level of consciousness about narcotic substances. The Sumerian clay tablet (about 2100 BC) is considered to be the world’s oldest recorded list of medical prescriptions. It is believed by some scholars that the opium poppy is referred to on the tablet [125]. Some objects from the ancient Greek Minoan culture may also suggest the knowledge of the poppy. A goddess from about 1500 BC is shown with her hair adorned probably with poppy capsules and her closed eyes insinuate sedation. Additionally, juglets probably imitate the poppy capsules found in that period in both Cyprus and Egypt. The first authentic reference to the milky juice of the poppy was found by Theophrastus at the beginning of the third century BC. In the first century, the opium poppy and opium were known by Dioscorides, Pliny, and Celsus, and later on by Galen. Celsus suggests the use of opium before surgery and Dioscorides recommended patients should take mandrake (which contains scopolamine and atropine) mixed with wine before limb amputation. The Arabic physicians used opium very extensively and in the 10th century; it was recommended by Avicenna, especially for diarrhea and diseases of the eye [125,126]. Polypharmacy, including a mixture of nonsensical medications, was often used. Fortunately, for both patients and physicians, many of the preparations contained opium. The goal was a panacea for all diseases. A famous and expensive panacea was theriaca, containing up to sixty drugs including opium. Simplified preparations of opium such as *Tinctura opii* were used up to about 2000 in Denmark. In the early 1800s, science had developed and Sertürner isolated morphine from opium and was the founder of alkaloid research. A safer and more standardized effect was obtained by pure opium [127].

Medical prescriptions for opioids started to increase in the 1990s, followed closely by significant increases in nonmedical use. Opioids are highly addictive, and the use of both synthetic and natural opioids can quickly result in dependence, which includes physical and/or psychological dependence, as well as opioid use disorder (OUD). OUD is a chronic relapsing disorder that, whilst initially driven by activation of brain reward neurocircuits, increasingly engages anti-reward neurocircuits that drive adverse emotional states and relapse, which is also related to dramatically increased rates of morbidity and mortality. An important risk factor for OUD and overdose death is the availability and volume of medical prescriptions for opioids [128,129]. Moreover, in some countries such as the USA, it is possible to buy opioids such as tramadol without a prescription on the internet [130]. A complex interplay of structural, social, developmental, and behavioral risk factors is likely to have a role in the development of OUD. An individual risk factor is male sex [131,132]; more men misuse and are addicted to opioids than women. However, clinical reports suggest that, for opioids, similar to other drugs of abuse, women progress from initial use to addiction at a faster rate than men [133]. Sex differences in the opioid system have been reported in preclinical studies, which might underlie sex differences in sensitivity to pain or addiction [134]. In addition, women have more acute pain, chronic pain such as dysmenorrhea, and chronic pelvic pain, and are prescribed opioids more often than their male counterparts. Studies have shown that there have been increased rates of use and overdose deaths in women and that significant mental health concerns are higher for them than in men [135].

Some studies have evaluated chronic opioid use and addiction in EM patients, for example, by conducting a cohort study between 2006 and 2017 comparing women aged 18–50 years with EM (N = 36,373) to those without (N = 2,172,936) in terms of risk of chronic opioid use, opioid dependence diagnosis, and opioid overdose [136]. Chronic opioid use was defined as ≥ 120 days’ supply dispensed or ≥ 10 fills of an opioid during any 365-day interval. EM patients had a four times greater risk of chronic opioid use compared to women without. Multimorbidity among these patients was associated with an elevated risk of chronic opioid use [136]. In another study, a retrospective cohort studied from 2011 to 2016 included 58,472 EM patients. Women who filled an opioid prescription within 12 months of diagnosis were placed in the opioid cohort and women who did not fill an opioid prescription were placed in the nonopioid cohort. Of these, 61.7% filled out an opioid prescription during the study period. More than 95% filled prescriptions for short-acting opioids (SAOs) only, 4.1% filled prescriptions for both SAOs and extended release/long-acting opioids (LAOs), and 0.6% filled prescriptions for LAOs only. Patients who filled an opioid prescription had higher baseline comorbidities (especially gynecologic and chronic pain comorbidities) and EM-related medication use compared with patients who did not fill out an opioid prescription. Patients who filled out both LAO and SAO prescriptions had the highest total day supply of opioids, the proportion of days covered by prescriptions, and morphine equivalent daily dose. These patients also had the highest proportions of opioid switching and dose augmentation in a retrospective analysis, with the study concluding that women with EM have higher probabilities of prolonged use of opioids and concomitant use with benzodiazepines compared with women without this condition [137,138]. An additional retrospective study from 2009 to 2018 evaluated all-cause and EM-related health care resource utilization and costs among newly diagnosed patients with high risk (≥1 day with ≥90 morphine milligram equivalents per day or ≥ 1 day concomitant benzodiazepine use) versus low risk opioid use or patients with chronic (≥90-day supply prescribed or ≥10 opioid prescriptions) versus non-chronic opioid use. Out of 61,019 patients identified, 18,239 had high risk opioid use and 5001 had chronic opioid use. The analysis demonstrated significantly higher all-cause and EM-related healthcare resource utilization and total costs for high risk opioid users compared to low risk opioid users among newly diagnosed EM patients over 1 year. Similar trends were observed comparing chronic opioid users with non-chronic opioid users, with the exception of EM-related pharmacy fills and associated costs [139].

Opioid addiction involves the hijacking of the endogenous opioid system [129]. The neurocircuitry of this addiction involves three stages: the intoxication stage (opioid intoxication and incentive salience), the withdrawal affect stage (opioid tolerance and withdrawal), and the anticipation stage (opioid craving and relapse) [129]. In the intoxication stage, preclinical studies reported that the reinforcing effects of opioids are mediated in the ventral tegmental area (VTA) and nucleus accumbens (NAc) via not only dopamine-dependent but also drug-independent mechanisms [140] (Figure 3). Neurobiological mechanisms of tolerance range from opioid receptor desensitization and downregulation to cellular and circuitry allostasis [141,142]. The anticipation stage of the addiction cycle involves dysfunction of executive function. In humans, opioid addiction has a dysregulated hypothalamic–pituitary–adrenal stress axis; this dysregulation persists during cycles of addiction [143,144]. MORs, DORs, and KORs play different roles in the mechanisms of opioid addiction: MORs promote recreational drugs and adapt to chronic activation, which also leads to tolerance and dependence; KORs enable and sustain aversive states of withdrawal and abstinence; and DORs are involved in the improvement in mood states and facilitate context learning. All of them modulate motivation. Both MOR and KOR activities drive the onset, progression, and maintenance of an addiction. Nevertheless, the contribution of DORs remains less straightforward [145].

More and more evidence supports the role of the N/OFQ–NOP system in addiction [146,147,148]. N/OFQ has a broad inhibitory effect on multiple neurotransmitter systems involved in drug reward and has been shown to decrease drug-induced dopamine levels in the nucleus accumbens [149,150]. Studies have shown that NOP agonists block the rewarding effects of morphine, cocaine, and alcohol in animal models of drug rewards such as the conditioned place preference (CPP) [148,151]. Furthermore, the fact that morphine-induced supraspinal analgesia and conditioned place preference are blocked by N/OFQ suggests that the N/OFQ–NOP system acts as an anti-opioid system for some responses. Targeting the NOP-N/OFQ system is therefore a potential approach to reduce the rewarding effects of multiple abused substances and develop pharmacotherapy to treat addiction to various drugs and possibly polydrug addiction. However, equivocal results with a few synthetic small-molecule NOP agonists were obtained [148,151]. Furthermore, in September 2021, Grünenthal announced that the first participants have been enrolled in a randomized, placebo- and active-controlled clinical trial for its peripherally restricted NOP receptor agonist. The compound is being developed to provide a non-opioid therapy option that offers a strong analgesic effect without the side effects commonly associated with opioids. The experimental medicine trial will evaluate the extent and duration of the pharmacological effect of the oral NOP agonist in an experimental pain model. The results of the trial were expected to be available in early 2022, but we could not find them until the publication of this paper [152].

## 7. Conclusions

Even though the pathophysiology of EM-associated pain is not completely understood, strong pieces of evidence support that neurogenic inflammation may play a crucial role. As we discussed above, current treatment options are not the best choices for EM-associated pain as they not only interfere with ovulation but also are ineffective for all patients. Endogenous opioid peptides can be secreted from immunocytes, occupy peripheral opioid receptors on sensory nerve endings, and produce analgesia by inhibiting the excitability of these nerves or the release of pro-inflammatory neuropeptides. For several hundreds of years, opium has been used for pain relief, and nowadays, opioids remain a gold standard for the treatment of pain. In some cases, it is recognized that there is no substitute for opioids in achieving satisfactory pain relief. However, undesirable side effects following acute administration and long-term use limit their clinical effect. Additionally, the already observed risk of chronic opioid use and addiction in EM patients suggests that the pharmacological drugs available now are not the best option for these patients. More and more studies have suggested that multi-mechanistic opioids can be a valid alternative to traditional opioids for their safer profile. Particular interest has been paid to the role of the NOP receptor in terms of analgesia and improved tolerability. Emerging evidence has shown a promising functional profile of bifunctional NOP/MOP partial agonists as safe and nonaddictive analgesics [85,89,153]. While coactivation of NOP and MOP receptors may provide a viable treatment option for pain and drug abuse, caution is warranted for bifunctional NOP/MOP “full” agonists. NOP/MOP “partial” agonists provide considerable hope for the future of NOP-active compounds and the potential for opioid-type analgesics with reduced side effects and abuse liability. However, until now, the role of NOP receptors in EM has not been investigated. Therefore, this review offers a thought that still needs further investigation but may provide potential options for relieving EM-associated pain.

## Figures and Tables

**Figure 1 ijms-24-01633-f001:**
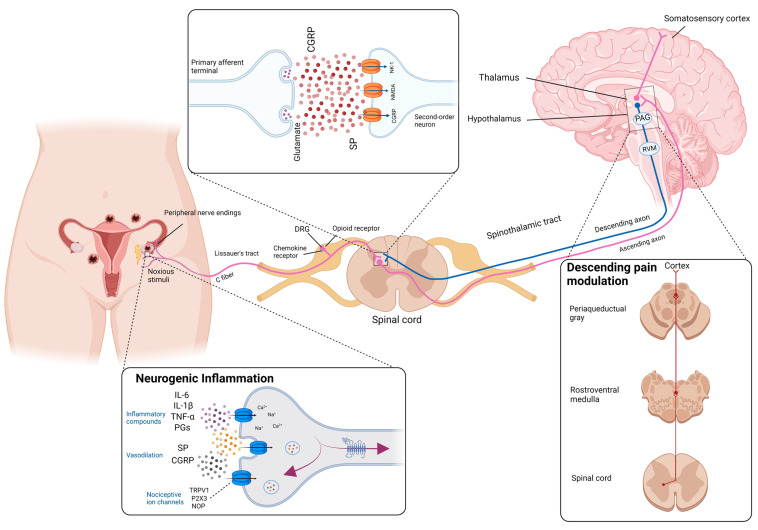
Pain and neurogenic inflammation in endometriosis. Endometriotic cytokines in the peritoneum, prostaglandins, interleukins (ILs), calcitonin gene-related peptide (CGRP), substance P (SP), nociceptive opioid peptide (NOP), and transient receptor potential vanilloid 1 (TRPV1) stimulate sensory peripheral nerve endings. The nociceptive signal conduct in the myelinated Aδ fibers or unmyelinated C fibers to the spinal cord. Subsequently, opioid peptides, which are synthesized as chemokine receptors, promote an antinociceptive effect in the dorsal root ganglia (DRG). In the DRG, chemokine receptors and opioid receptors receive the nociceptive message and transfer it to the spinal cord. Within the spinal cord, neurotransmitters such as CGRP, SP, and glutamate are released to activate the second-order neurons. Then, CGRP will combine with the CGRP receptor complex and SP will act on the neurokine-1 receptor (NK-1). The noxious message travels up to the thalamus via ascending axons and then finally reaches the somatosensory cortex. The nociceptive input can be attenuated or facilitated through endogenous opioid release via a descending pain-modulated system, which includes two crucial structures, periaqueductal gray (PAG) and rostroventral medulla (RVM).

**Figure 2 ijms-24-01633-f002:**
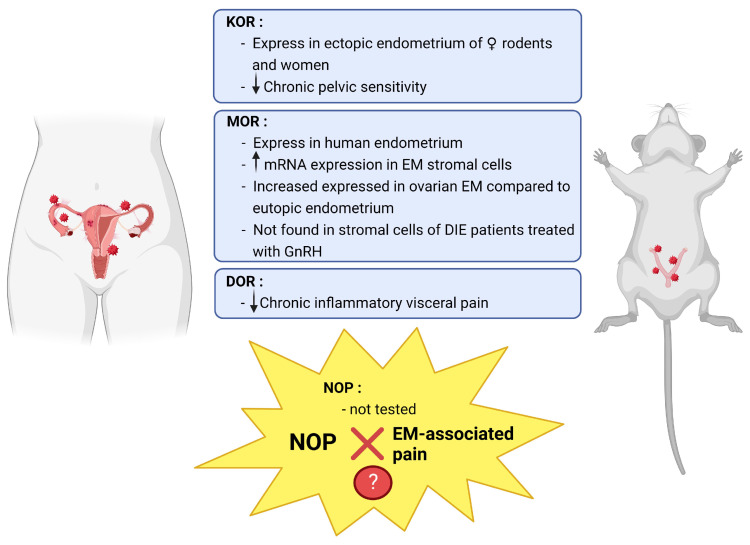
Opioid receptors and EM-associated pain. The endogenous opioid system is involved in modulating neurogenic inflammation, while endometriosis is a chronic inflammatory disease. In the studies regarding opioid receptor expression in the endometrium, both MOR and KOR are more or less related to EM-associated pain. Although to date, no evidence has shown that DOR antagonists can alleviate EM-associated pain, DOR agonists were reported to be active in the periphery and probably alleviate the chronic inflammatory visceral pain. Regarding NOP, what we know is its wide expression both in the central and peripheral nervous systems. Meanwhile, NOP receptors also have a high distribution within the immune system. To date, the relationship between EM-associated pain and NOP or N/OFQ remains unknown. KOR: kappa opioid receptors; MOR: mu-opioid receptors; DOR: delta opioid receptors; NOP: nociceptin/orphanin FQ peptide receptor; EM: endometriosis; GnRH: gonadotropin-releasing-hormone; DIE: deep infiltrating endometriosis.

**Figure 3 ijms-24-01633-f003:**
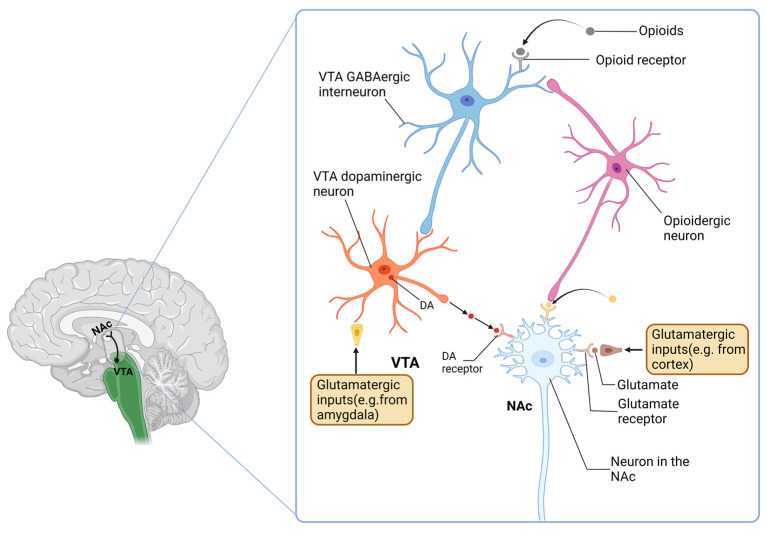
Rewarding actions of opioids in the VTA and NAc. Drug abuse, especially opioid abuse, has some significant effects on the area of VTA and NAc. γ -aminobutyric acid (GABA)ergic interneurons in the VTA are inhibited by opioids mainly via μ-opioid receptors but also by δ-opioid receptors. This leads to the disinhibition of VTA dopaminergic neurons and activation of reward circuitry in the Nac. Furthermore, reward circuitry is also activated directly by opioids through opioid receptors on NAc neurons. VTA: ventral tegmental area; NAc: nucleus accumbens.

## Data Availability

Not applicable.

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
