# Peer review of "Endometriosis and Opioid Receptors: Are Opioids a Possible/Promising Treatment for Endometriosis?"

_ijms, 2023, doi:10.3390/ijms24021633_

Round 1
Reviewer 1 Report
The manuscript " Endometriosis and Opioid receptors: Are opioids a possible/good treatment for endometriosis?" was found to be of potential significance. The discussion is interesting and relevant to the field, especially endometriosis-related pain which is critically imperative for endometriosis patients. Nevertheless, there are some comments and suggestions listed below for the authors.
Title:
Please change it to: "Endometriosis and Opioid receptors: Are opioids a possible/promising treatment for endometriosis-related pain?”
Abstract:
The overall conclusion of the study should be summarized and added to this section.
Keywords:
Try to avoid repeating words used in the title. Keywords are used in databases to search for research; using different words increases the probability of your work being found.
Introduction:
The introduction does not provide a clear rationale concerning the goals and hypotheses of the study. Please provide more details on the main aims of the study and a short background in the introduction section.
Body:
It would be helpful to add a separate section devoted to how analgesics, especially opioids, affect infertility and reproduction in endometriosis patients and animal models.
Conclusion:
Even though the different parts of the study have been well summarized, a more accurate and specific conclusion could enhance the quality of the study.
Reviewer 2 Report
The authors focused on Endometriosis and Opioid receptors. It was well written and explained review.
However, I would like to suggest adding the current efforts and challenges in developing opioids and how they can be a better target in the treatment of endometriosis would be of great value to this Review.
Adding on opioid addiction would also further enhance this review
